# Should Degenerated Intervertebral Discs of Patients with Modic Type 1 Changes Be Treated with Mesenchymal Stem Cells?

**DOI:** 10.3390/ijms23052721

**Published:** 2022-02-28

**Authors:** Nick Herger, Paola Bermudez-Lekerika, Mazda Farshad, Christoph E. Albers, Oliver Distler, Benjamin Gantenbein, Stefan Dudli

**Affiliations:** 1Center of Experimental Rheumatology, University Hospital Zurich and Balgrist University Hospital, University of Zurich, CH-8008 Zurich, Switzerland; nick.herger@usz.ch (N.H.); oliver.distler@usz.ch (O.D.); 2Tissue Engineering for Orthopaedics and Mechanobiology, Bone & Joint Program, Department for BioMedical Research (DBMR), Medical Faculty, University of Bern, CH-3008 Bern, Switzerland; paola.bermudez@dbmr.unibe.ch (P.B.-L.); benjamin.gantenbein@dbmr.unibe.ch (B.G.); 3Department of Orthopaedic Surgery and Traumatology, Inselspital, Bern University Hospital, Medical Faculty, University of Bern, CH-3010 Bern, Switzerland; christoph.albers@insel.ch; 4Department of Orthopaedics, Balgrist University Hospital, CH-8008 Zurich, Switzerland; mazda.farshad@balgrist.ch

**Keywords:** mesenchymal stem cell, stem cell therapy, Modic change, intervertebral disc, regeneration, immunomodulation

## Abstract

Low back pain (LBP) has been among the leading causes of disability for the past 30 years. This highlights the need for improvement in LBP management. Many clinical trials focus on developing treatments against degenerative disc disease (DDD). The multifactorial etiology of DDD and associated risk factors lead to a heterogeneous patient population. It comes as no surprise that the outcomes of clinical trials on intradiscal mesenchymal stem cell (MSC) injections for patients with DDD are inconsistent. Intradiscal MSC injections have demonstrated substantial pain relief and significant disability-related improvements, yet they have failed to regenerate the intervertebral disc (IVD). Increasing evidence suggests that the positive outcomes in clinical trials might be attributed to the immunomodulatory potential of MSCs rather than to their regenerative properties. Therefore, patient stratification for inflammatory DDD phenotypes may (i) better serve the mechanisms of action of MSCs and (ii) increase the treatment effect. Modic type 1 changes—pathologic inflammatory, fibrotic changes in the vertebral bone marrow—are frequently observed adjacent to degenerated IVDs in chronic LBP patients and represent a clinically distinct subpopulation of patients with DDD. This review discusses whether degenerated IVDs of patients with Modic type 1 changes should be treated with an intradiscal MSC injection.

## 1. Introduction

A fundamental challenge for improving the lives of chronic low back pain (CLBP) patients is the lack of effective targeted treatments. The development of novel targeted therapies for CLBP patients is hampered in part by the heterogeneity of the CLBP population. Pain may arise from several anatomical structures, including the intervertebral disc (IVD), the endplate, the vertebral body, the facet joints, the spinal ligaments, and the muscles. Central pain sensitization and psychosocial factors can further complicate the diagnosis.

Degenerative disc disease (DDD) is one of the most common findings in CLBP patients. DDD is an inflammatory–catabolic process triggered by a long list of genetic, mechanical, and environmental factors that ultimately leads to the resorption of the IVD. Anti-inflammatory and regenerative approaches have been attempted to treat degenerated discs. In the past 15 years, many cell therapy approaches for DDD have been developed, several of which have reached phase I and II clinical trials, and a few phase III trials [1].

Patient stratification is critical for showing a clinically meaningful treatment effect. However, the high prevalence of disc degeneration (DD) in the heterogeneous CLBP population [2] and the high percentage of asymptomatic individuals with DD (31.5–37.5%) [3] limit the sensitivity and specificity of DD for CLBP and hence impose a major challenge to stratify patients for a potential therapy.

Modic changes (MC) are vertebral bone marrow lesions that are almost exclusively present at levels with DD. MC are frequently observed in CLBP patients. A systematic review investigated the prevalence of MC and reported a median prevalence of 43% in CLBP patients and 6% in a non-clinical population [4]. Prevalence generally increases with age and peaks in the 60s [5]. Accumulating evidence shows that CLBP patients with DDD and MC are different from DDD patients without MC [6]. Patients with MC report a greater frequency and duration of low back pain (LBP) episodes, seek care more often, have a higher risk of a poor outcome, and have an ‘inflammatory pain pattern’ [4,7,8,9,10,11]. Larger lesions seem more painful and have a positive predictive value for pain of up to 100% [12,13]. Therefore, MC patients may in fact represent a clearly defined subpopulation of DDD patients. However, the effects of discal cell therapy at spinal levels of MC remain unknown. Thus, we discuss in this review whether patients with MC should be considered for a specific discal cell therapy.

### Modic Changes

Vertebral bone marrow lesions adjacent to degenerated discs were first described by Assheuer et al. in 1987 [14] and later coined by Modic et al. in 1988 [15]. Three interconvertible types of MC have been defined based on their appearance in T1-weighted and T2-weighted magnetic resonance imaging (MRI) (Figure 1) [15,16]. Histological data of MC patient bone marrow are sparse [14,15,17,18]. In Modic type 1 changes (MC1), fibrosis, granulation tissue, lymphocytic and neutrophilic infiltrations, increased frequency of adipocytes, necrotic adipocytes, and interstitial water have been reported [14,15,19]. In Modic type 2 changes (MC2), the red hematopoietic bone marrow is replaced by fatty bone marrow and can contain displaced disc tissue along with fibrotic tissue [14,15,20]. Trabecular bone in MC1 is thinned, possibly due to osteoclastic activity, and thickened in MC2 [14,15,17]. Modic type 3 changes (MC3) represent extensive sclerotic changes [15,17]. Increased numbers of peptidergic nerve endings were found in MC1 and to a lesser extent in MC2 [18,19]. This may relate to the high specificity of MC for pain in discography [13,20].

The IVD and the vertebral endplate seem to play an important role in the pathomechanism of MC (Figure 2). MC only occur adjacent to degenerated discs and mostly develop simultaneously in the cranial and caudal vertebrae of the degenerated disc [16]. Progression of DD accompanies the progression or evolution of MC [21]. Vertebral endplate defects are strongly associated with MC and extensive endplate degeneration is a risk factor for the progression of DD and MC [21,22]. Endplate defects enhance the fluid flow between the disc and the bone marrow [23,24] and may provide a physical explanation for the inflammatory and pro-fibrotic cross-talk between the disc and the bone marrow observed in MC [25]. This cross-talk likely promotes MC development, thus representing an interesting treatment target. In vivo studies with mice, rats, and baboons confirm that disc injury can cause changes in the adjacent vertebrae, with alterations in marrow composition and remodeling of trabecular bone [26,27,28,29]. Analysis of human disc samples revealed increased expression of pro-inflammatory, pro-osteoclastic, and neurotrophic cytokines (Table 1) [19,30,31]. Notably, many of them can affect hematopoiesis and contribute to the hematopoietic changes observed in MC bone marrow [25].

Despite an increasing understanding of the molecular and cellular changes in MC bone marrow and discs, the etiology of MC remains largely unknown. Autoinflammation against disc material and occult disc infection are both supported by clinical and experimental studies [30,31,32,33,34,35,36,37]. While infectious MC may be treated with antibiotics [36,37], no approved treatment or treatment consensus exists for autoinflammatory MC. Standard treatments for CLBP are generally less effective in MC1 patients [8,38]. Treatment attempts with intradiscal steroids, bisphosphonates, and tumor necrosis factor alpha (TNF-α) inhibitors to control inflammation had limited short-term efficacy, and lacked anatomical or biological specificity [39,40,41,42,43,44]. Spinal fusion surgery may relieve pain but can have serious risks besides those of surgery and anesthesia [45].

In summary, clinical and experimental data suggest that disc inflammation can affect the adjacent bone marrow via an enhanced cross-talk through damaged endplates. Therefore, suppression of discal inflammation might represent a promising treatment strategy to protect the bone marrow from the vicious cross-talk with the inflammatory disc.

**Figure 2 ijms-23-02721-f002:**
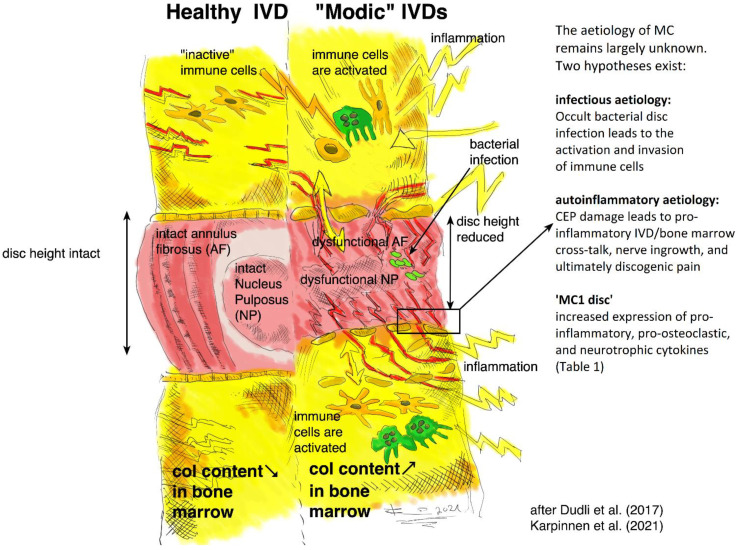
Schematic illustration of possible causes of pain and inflammation in ‘MC discs’, comparing a ‘healthy disc’ (on the left side) to a ‘Modic disc’ (on the right side). Note that the central role is given to the CEP: CEP damage possibly enables inflammation in the adjacent vertebrae, triggering a cross-talk to inflammatory cells. Ingrowth of nerve endings into the IVD might be responsible for pain development. Increased osteoclast activity might be responsible for the inflammatory trabecular bone resorption observed in MC1 [14,15,17]. MSCs in the bone marrow adjacent to ‘MC1 discs’ have a pro-fibrotic phenotype [46], possibly due to the pro-fibrotic and pro-inflammatory cross-talk with the ‘MC1 disc’.

## 2. Clinical Trials for MC

A literature review was carried out in January 2022 across the ClinicalTrials.gov (accessed on 22 February 2022) database. Keywords used in the selection of clinical trials were: “modic”, “discopathy”, and “endplate changes”. All pharmaceutical interventional studies were considered and their relevance to the subject of this review was checked. Multiple clinical trials investigating the efficacy of antibiotics or intradiscal steroid injections for the treatment of CLBP patients with MC were not listed in the ClinicalTrials.gov (accessed on 22 February 2022) database. Eligible studies were selected and are summarized in Table 2.

Various drugs have been tested in clinical trials to treat CLBP patients with and without MC. The AIM study investigated the use of amoxicillin to treat occult disc infections in a double-blind, randomized, phase III, placebo-controlled, multicenter clinical trial (NCT02323412) [47,48]. The study showed that amoxicillin failed to provide a clinically meaningful benefit for CLBP patients with MC. Therefore, the study results did not support the use of amoxicillin for CLBP patients with MC. Interestingly, when the analysis was limited to the inflammatory MC1, they found significant improvements in pain and disability. However, the minor clinical benefit failed to reach a predefined threshold for clinically meaningful improvements. The treatment efficacy might have been limited to a group of responders, which would explain the weak overall efficacy. On the contrary, two randomized, placebo-controlled studies investigated the use of amoxicillin and clavulanate for the management of CLBP in patients with disc herniation and MC1 (NCT00302796) [49,50]. They found significant pain and disability-related improvements in the cohort treated with antibiotics compared to the placebo cohort. A more recent prospective, open-label study used the same treatment regimen yet failed to replicate the treatment outcomes [51]. In summary, studies using antibiotics to treat CLBP patients with MC are inconsistent. A potential explanation for these inconsistent study results might be the diverse percentages of patients with infectious MC between the different trials. While patients with infectious MC represent a valid target population, those with autoinflammatory MC might not benefit from antibiotics. This highlights the need for new diagnostics that enable the stratification of the MC study population into infectious and autoinflammatory subpopulations.

Bisphosphonates might consolidate inflamed vertebral bodies, thereby improving the tolerance for mechanical load. A phase II, randomized, placebo-controlled, double-blinded clinical trial investigated the efficacy of zoledronic acid (ZA) for CLBP patients with MC (NCT01330238) [42,43]. Besides a higher frequency of adverse events in the ZA group compared to the placebo group, this trial reported minimal pain and disability-related improvements after one month in favor of the ZA group. No significant difference in pain and disability-related improvements was observed after one year between the ZA group and the placebo group. Interestingly, MC1 volumes tended to decrease in the ZA group, but increased in the placebo group. Furthermore, MC1 to MC2 conversion was more common in the ZA group, although statistical significance was not demonstrated. In summary, ZA tended to accelerate the conversion from MC1 to less painful MC2. Even though the study results are encouraging, larger studies are required to verify the efficacy of ZA in the treatment of CLBP patients with MC.

Glucocorticoids are potent anti-inflammatory and immunosuppressive drugs, frequently used to treat various inflammatory, allergic, and autoimmune disorders. Clinical trials investigated the efficacy of intradiscal steroid injection in a heterogeneous study population of patients suffering from discogenic LBP. These trials reported no significant clinical benefit in the use of intradiscal steroids [52,53]. More recent clinical trials on intradiscal or epidural steroid injections recognized the importance of stratifying CLBP patients and focused specifically on CLBP patients with MC. The reported outcomes on intradiscal steroid injections across multiple clinical trials [39,40,41,54,55] and case reports [56,57] are consistent. Rapid pain- and disability-related improvements could be found but the long-term efficacy of intradiscal and epidural steroid injections could not be demonstrated. This reinforces the hypothesis that MC is a type of inflammation that can and should be treated. Glucocorticoids might not be the right treatment approach to target the discal inflammation because they failed to modify the disease, as indicated by the recurrence of pain after intradiscal and epidural steroid injection.

The BackToBasic study is currently recruiting CLBP patients with MC1 to compare the effect of Infliximab—an intravenously administered TNF-α inhibitor—to a placebo (NCT03704363) [58]. This pharmaceutical intervention aims to target the underlying discal inflammation in MC1.

Antibiotics to treat occult disc infections, anti-inflammatory drugs to suppress discal inflammation, and bisphosphonates to consolidate vertebral bodies all represent valid causal treatment approaches. Current clinical trials seem to have recognized the pivotal role of inflammation in MC1 and therefore are looking into anti-inflammatory treatment approaches.

A recent review highlighted the central role of inflammation in DDD independent of the presence of MC [59]. The review summarizes outcomes from anti-inflammatory clinical interventions in discogenic LBP patients [59]. TNF-α inhibitors were previously shown to be effective in the treatment of sciatica (NCT00364572) [60]. An open-label study used Infliximab to treat disc herniation-induced severe sciatica and reported promising early results [61,62]. This study was followed up by a randomized, placebo-controlled trial (FIRST II) to validate these early results [44,63]. Unfortunately, the study results did not support the use of Infliximab for the treatment of patients with herniation-induced sciatica. Interestingly, they showed that Infliximab treatment was beneficial if MC were present adjacent to the symptomatic level. This finding might have laid the foundation for the ongoing BackToBasic study that uses Infliximab for CLBP patients with MC1 [58]. Tanezumab—a potent anti-nerve growth factor (NGF) antibody—was used in several studies on CLBP (NCT00584870 [64], NCT00876187 [64,65], NCT00924664 [66,67]). Overall, tanezumab treatment led to a significant LBP reduction. However, adverse events including abnormal peripheral sensation, arthralgia, and accelerated osteoarthritis progression in affected patients were observed [64,67]. Platelet-rich plasma (PRP) represents a multimodal treatment approach, as it contains growth factors, blood-clotting factors, proteinase inhibitors, and immunomodulatory factors. Completed studies on PRP for CLBP patients consistently reported significant pain relief and disability-related improvements [68,69,70]. The promising study results might be attributed to the multimodal effects of PRP, including the reduction of inflammation and promotion of tissue regeneration [59]. Since MC1 are highly associated with DD and inflammation, it might be insightful to consider these study outcomes for future clinical trials on anti-inflammatory interventions for DDD patients with MC1.

**Table 2 ijms-23-02721-t002:** Summary of pharmaceutical interventional clinical trials on the treatment of CLBP patients with MC.

Clinical Trial/Study	Year	Aim	Treatment	Phase and Design	Inclusion Criteria	Number of Patients	Status and Outcome	Outcome Measures	References
Antibiotics in Modic changes (AIM) NCT02323412	2015 2018	Effects of amoxicillin in CLBP patients with MC at the disc herniation level	Amoxicillin 750 mg three times a day for three months	Phase III. Double-blind, multicenter, randomized, placebo-controlled	CLBP patients with disc herniation and MC1 and/or MC2 at the same level	180	Completed Without clinically important benefit	- LBP intensity scores - Oswestry Disability Index - Roland-Morris Disability Questionnaire	[47,48]
Antibiotic treatment of patients with low back pain NCT00302796	2006 2010	Effect of antibiotics in CLBP patients with MC1	Amoxicillin-clavulanate (500/125 mg) three times a day for 100 days	Phase IV. Double-blind, randomized, placebo-controlled	CLBP patients with disc herniation and MC1	162	Completed Clinically important benefit	- LBP intensity scores - Disease-specific disability - Global perceived health - MRI	[49]
Antibiotic treatment for the management of CLBP ACTRN12615000958583	2015	Efficacy of antibiotics in a broader subgroup of CLBP patients with disc herniation	Amoxicillin-clavulanate (500/125 mg) two times per day for 90 days	Double-blind, randomized, placebo-controlled	CLBP patients with disc herniation—with and without MC1 and MC2	170	Recruiting	- LBP intensity scores - Self-reported disability - Work absenteeism - Hindrance in work performance	[71]
Antibiotic treatment of CLBP patients with MC1	2016	Efficacy of antibiotic treatment of CLBP patients with MC1	Amoxicillin-clavulanate (500/125 mg) three times a day for 100 days	Prospective, open-label	CLBP patients with MC1	28	Completed No clinically important benefit	- LBP intensity scores - Self-reported improvement - Analgesics consumption - Spinal steroid injections	[51]
Antibiotics in CLBP patients with MC1	2014	Efficacy of antibiotics in CLBP patients with MC1	Amoxicillin-clavulanate (500/125 mg) two times per day for 100 days	Randomized, placebo-controlled	CLBP patients with disc herniation and MC1	71	Completed Clinically important benefit	- LBP intensity score - Roland-Morris Disability Questionnaire	[50]
PP353 for CLBP patients with MC1 NCT04238676	2020	Safety, tolerability, and efficacy of PP353	Intradiscal injection of the antibiotic PP353	Phase I/II. Randomized, placebo-controlled	CLBP patients with MC1	43	Recruiting	- Adverse events incidence - LBP intensity scores - Roland-Morris Disability Questionnaire	-
The efficacy of Zoledronic Acid in MC-related LBP NCT01330238	2008 2011	Efficacy of Zoledronic Acid in patients with CLBP and MC	Single infusion of 5 mg zoledronic acid	Phase II. Double-blind, randomized, placebo-controlled	CLBP patients with MC1 or MC2	40	Completed Reduced LBP and faster MC1 conversion to MC2	- LBP intensity scores - Oswestry Disability Index	[42,43]
Intradiscal steroid injection in CLBP with inflammatory MC	2007	Association between MC severity and response to intradiscal steroid injection	Intradiscal injection of 25 mg prednisolone acetate	Retrospective	CLBP patients with MC1, MC1/2, or MC2	74	Completed Significant short-term benefit	- LBP intensity score	[39]
Intradiscal steroid therapy for CLBP patients with MC	2011	Efficacy of various intradiscal steroid injection regimens for CLBP patients with MC	Intradiscal injection of normal saline, disprospan, or disprospan and songmeile	Double-blinded, randomized, placebo-controlled, prospective	CLBP patients with MC and positive discography	120	Completed Significant short-term benefit	- LBP intensity score - Oswestry Disability Index	[40]
Intradiscal steroid injection in CLBP patients with MC1	2012	Efficacy of intradiscal steroid injection on CLBP patients with MC1	Intradiscal injection of methylprednisolone	Retrospective	CLBP patients with and without MC1	97	Completed Significant short-term benefit	- Self-reported improvement	[41]
Intradiscal glucocorticoid injection for CLBP patients with active discopathy NCT00804531	2017	Efficacy of single intradiscal glucocorticoid injection in CLBP patients and active discopathy	Single intradiscal injection of 25 mg prednisolone acetate	Phase IV. Prospective, parallel-group, double-blind, randomized, placebo-controlled	CLBP patients with active discopathy	135	Completed Significant short-term benefit with no long-term benefit	- LBP intensity score - MRI - Disability - Quality of life - Use of analgesics	[54]
Epidural steroid injections in discogenic LBP NCT04930211	2020	Effectiveness of epidural steroid injections in DDD patients with/without MC1	Transforaminal epidural steroid injection of dexamethasone-lidocaine	Non-randomized without placebo	CLBP patients with/without MC1	40	Recruiting	- LBP intensity score - Oswestry Disability Index	-
BackToBasic: Infliximab in CLBP and MCs NCT03704363	2018	Efficacy of Infliximab in CLBP with MCs	Four intravenous Infliximab infusions (5 mg/kg)	Phase III. Double-blind, multicenter, randomized, placebo-controlled	CLBP patients with MC1	126	Recruiting	- LBP intensity score - Oswestry Disability Index - Incidence of adverse events - Roland-Morris Disability Questionnaire	[58]
Intradiscal injection of PRP for CLBP patients with MC1 NCT03712527	2018	Efficacy of intradiscal PRP injection at 3 months	Single intradiscal PRP injection versus normal saline	Randomized, placebo-controlled	Patients with at least 3 months LBP with MC1	126	Recruiting	- LBP intensity score - Roland-Morris Disability Questionnaire - Analgesics consumption	-

## 3. MSC Therapy for DDD

Mesenchymal stem cells (MSCs) have been used in numerous clinical trials for DDD independent of concomitant MC during the past decade. Clinical trials for DDD using MSCs have recently been reviewed [1]. MSCs are an attractive option for cell therapies because of their self-renewal capacity, ease of isolation, multi-lineage differentiation potential, engraftment capacity, safety profile, and immunomodulatory properties. MSC treatments have been shown to be safe and well-tolerated, with no reported severe adverse events but with occasional mild pain-related adverse events [1,72,73]. Previous studies observed no host immune rejection against allogeneic MSCs, indicating that allogeneic MSCs might avoid immunogenic reactions in humans [1,74,75]. Although concerns about MSC-associated tumorigenesis, osteophyte formation, infection, and immune rejections are justified, none of these safety concerns have been confirmed in completed clinical trials on MSC injections in DDD patients.

The underlying concept of regenerative treatment approaches in DDD assumes that MSC injection reconstitutes the healthy disc anatomy and thereby restores the normal functioning of the motion segment [76]. The successful restoration of a functionally impaired motion segment could prevent the development of degenerative spinal pathologies [72,73,76]. The chance to disrupt the degenerative cascade led to the conduction of multiple clinical trials, investigating the use of intradiscal MSC injections to treat DDD [1]. Most completed clinical trials focused on IVD regeneration and the safety profile of MSC injections. Additionally, pain relief and disability-related improvements have been frequently investigated yet controlling disc inflammation has not been a focus of these trials. Despite tremendous efforts, substantial IVD regeneration could not be shown in any of the completed clinical trials on intradiscal MSC injections in DDD patients, besides occasional IVD rehydration and deceleration of degenerative processes [1]. Interestingly, DDD patients treated with intradiscal MSC injections experienced substantial pain relief and showed significant disability-related improvements in a group of responders [1,75,77]. These findings raised the question of whether IVD regeneration is essential for achieving favorable therapy outcomes and indicated that the analgesic effect of the MSC injections may be due to a regeneration-independent mode of action.

The multifactorial etiology of DDD could manifest in various sources of discogenic pain. The key processes of DDD that are related to discogenic pain have been reviewed elsewhere [74]. In summary, inflammation, an acidic IVD microenvironment, nerve ingrowth, and endplate damage were found to be closely linked to discogenic pain in DDD [59,74,75,77,78,79,80]. All these features are characteristics of MC, making MC one of the DDD-associated findings with the highest pain specificity [20,81]. These sources of discogenic LBP could potentially be targeted by the broad mode of action of MSCs, including the secretion of immunomodulatory factors, multi-lineage differentiation potential, and the promotion of cell survival [82,83].

In order to assess whether MSCs should be considered for ‘MC discs’, we next review the regenerative and immunomodulatory mode of action of MSCs and their contribution to pain relief.

### 3.1. Regenerative Mode of Action

Structural damage of the IVD and cartilage endplate (CEP) might contribute to functional impairment of the vertebral motion segment, which in turn could result in painful discal inflammation [73,76]. Targeting the underlying biomechanical issue might alleviate or even prevent painful vertebral bone marrow inflammation and subsequent degenerative processes. MSCs can promote IVD regeneration by various mechanisms. Their ability to proliferate and differentiate into chondrocytes [84,85,86,87,88] could allow them to replace damaged IVD cells (IVDCs), thereby supporting chondrogenesis [89]. Additionally, MSCs can contribute to IVD regeneration by the de novo synthesis of the extracellular matrix (ECM) [90]. Paracrine secretion of anabolic growth factors, anti-catabolic factors, and immunomodulatory cytokines by MSCs influences the survival and function of resident IVDCs [90] and renders MSCs promising candidates for inducing IVD regeneration [85,90,91,92,93,94].

The cross-talk between MSCs and IVDCs was shown to downregulate the gene expression of pro-inflammatory cytokines in IVDCs and to significantly increase their insoluble collagen synthesis and proliferation rate in vitro [92]. Furthermore, the co-culture of MSCs with nucleus pulposus cells (NPCs) was shown to protect NPCs against compression-induced apoptosis by reducing the concentration of reactive oxygen species and maintaining mitochondrial integrity [95]. On the contrary, the cross-talk between MSCs and IVDCs did not lead to the significantly increased synthesis of insoluble collagen by MSCs, but clearly induced the gene expression of various growth factors [92]. The paracrine secretion of these growth factors might have led to the increased proliferation and collagen synthesis of IVDCs.

Unfortunately, in vitro studies are incapable of mimicking the complex IVD microenvironment. The hostile IVD microenvironment likely impairs the regenerative potential of MSCs [77]. To enable MSCs to contribute to substantial IVD regeneration, they must be adapted to or protected from the harsh IVD microenvironment, including nutrient deprivation, hypoxia, acidic pH, high osmolarity, and a combination of inflammatory cytokines [60,86,96,97]. The use of MSC licensing strategies and biomaterial scaffolds may help to improve the survivability and the therapeutic potential of MSCs. Modulation of the inflammatory environment before attempting to regenerate the IVD and CEP using MSCs might be necessary. A non-inflammatory IVD microenvironment is more likely to support larger numbers of chondrogenic MSCs, which are needed to induce regeneration [77,98].

The following section summarizes the characteristics of the IVD microenvironment, which were discussed in a comprehensive review by Vadalà et al. [99]. We review the impact of the harsh IVD microenvironment on IVDCs and MSCs and discuss its relevance in MC.

#### 3.1.1. Nutrient and Oxygen Deficiency

The avascular nature of IVDs creates a hypoxic microenvironment with limited nutrient availability. Hypoxia (2–5% O_2_) and low glucose (1 mg/mL) were found to have positive effects on MSC-mediated IVD regeneration. MSCs cultured under hypoxia not only grew significantly faster than MSCs cultured under normoxia, but also had increased expression of genes associated with ECM assembly and improved differentiation potential [96,100,101,102,103]. The viability and proliferation of MSCs were maintained at IVD-like low glucose levels, whilst ECM biosynthesis was significantly enhanced [97,104]. Similarly, hypoxia supports the survival of NPCs and significantly enhances ECM biosynthesis and NPC proliferation [105,106,107,108]. Nutrients and oxygen are transported in blood vessels to the CEP and small capillaries supply nutrients through the CEP to the CEP/annulus fibrosus (AF) interface [109]. The nutrient supply of IVDCs then depends on diffusion from the CEP/AF interface into the IVD [109]. Endplate calcification increases with ageing and progression of DD and likely limits the diffusion of nutrients as the capillaries can no longer penetrate the endplate or are damaged as a result of the calcification [25,97,110]. Endplate defects are frequently seen in MC and have been shown to be responsible for substantial changes in diffusion between the IVDs and adjacent vertebral bodies [24]. No data on nutrient and oxygen concentrations in ‘MC discs’ have been published, but it can be speculated that the increased diffusion through damaged endplates increases the nutrient concentration and oxygen tension in ‘MC discs’, with unknown consequences for IVDC behavior and intradiscally injected MSCs.

#### 3.1.2. Acidity

Non-degenerated IVDs have a pH between 7.1 and 7.4 but the pH can drop to 6.8 in mild DD and can reach values of 6.2 in severe DD [111,112]. The proliferation rate and viability of MSCs and NPCs decrease with increasing acidity [113]. Furthermore, an acidic pH stimulates NPCs to increase the secretion of pro-inflammatory cytokines, nerve growth factors, and catabolic enzymes [114,115,116]. The pH in ‘MC1 discs’ has not yet been investigated. Endplate leakage enhances the fluid flow between the IVD and the bone marrow, thus likely facilitating the efflux of acidic metabolites into the adjacent bone marrow. A low pH can lower the threshold for the activation of sensory nerve fibers through acid-sensing sodium channels and is hence directly linked to nociceptive pain [117]. This might be relevant in MC, because more sensory nerve fibers were found in the bone marrow close to the endplates [18,19].

#### 3.1.3. Hyperosmolarity

The IVD has a hyperosmolar environment. In non-degenerated IVDs, osmolarity ranges between 430 and 500 mOsm/L [118] but steadily declines with the progression of DD due to loss of proteoglycans [118]. A hyperosmolar culture condition (485 mOsm/L) significantly decreases the gene expression of aggrecan and collagen-1 and decelerates MSC proliferation compared to standard cell culture conditions (280 mOsm/L) [104]. Therefore, reduced hyperosmolarity in degenerated IVDs might be beneficial for ECM deposition and the proliferation rate of intradiscally injected MSCs. Osmolarity in ‘MC discs’ has not been investigated; thus, the effect of osmolarity on intradiscally injected MSCs in ‘MC discs’ remains unknown.

In summary, it is challenging to restore a healthy IVD microenvironment by addressing single components of the complex network of cytokines, growth factors, catabolic enzymes, and neurotrophic factors found in degenerated IVDs [119]. An adaptable multimodal therapeutic approach might be needed to suppress the discal inflammation and to restore a healthy IVD microenvironment.

### 3.2. Immunomodulatory Mode of Action

The exceptional potential of MSCs to modulate a broad range of immune cells makes them an interesting candidate for the treatment of inflammatory disorders. The impact of MSCs on the functional properties of various cells from the innate and adaptive immune system has been thoroughly reviewed elsewhere [110,120]. In summary, MSCs can modulate immune cells through a paracrine mode of action. The secretion of immunomodulatory factors, including indoleamine 2,3-dioxygenase (IDO), TNF-α-inducible protein 6 (TSG-6), PGE2, IL-10, and transforming growth factor-beta (TGF-β), has distinct effects on various cell types. MSCs can regulate the antibody secretion of B cells, suppress T cell activation and proliferation, and prevent the activation of neutrophils [120]. Furthermore, MSCs can inhibit the maturation of dendritic cells and polarize macrophages towards immunomodulatory M2 macrophages. It has been shown that macrophages and other leukocytes [119] can infiltrate contained IVDs during degeneration and the number of infiltrated macrophages positively correlated with the progression of DD [114,121]. This might indicate the importance of MSCs in modulating a broad range of immune cells to resolve discal inflammation. Besides the secretion of immunomodulatory factors, MSCs interact with immune cells via cell–cell contact, mitochondrial transfer, and extracellular vesicles [1,110,115,116,122]. Regarding intradiscal MSC therapy, it is important that the immunomodulatory action of MSCs is not limited to leukocytes but also affects disc cells.

The secretion of inflammatory factors by IVD cells and infiltrating immune cells not only shifts the balance between anabolic and catabolic processes towards ECM degradation but also promotes the secretion of neurotrophic factors by IVD cells, ultimately leading to nerve ingrowth and discogenic pain [59,119,121,123,124,125,126]. Interestingly, co-culture of MSCs with degenerative IVDCs significantly downregulated the gene expression of pro-inflammatory cytokines (interleukin-1α (IL-1α), IL-1β, IL-6, TNF-α) [92]. Moreover, the co-culture of MSCs with degenerative NPCs significantly upregulated the gene expression of various growth factors (epidermal growth factor (EGF), insulin-like growth factor-1 (IGF-1), osteogenic protein-1 (OP-1), growth differentiation factor-7 (GDF-7), and TGF-β in MSCs. This study demonstrated the immunomodulatory potential of MSCs to modulate the degenerative IVDCs. To consider MSCs for the treatment of MC, MSCs must suppress the inflammation in the ‘MC disc’. Elevated levels of inflammatory molecules including TNF-α, IL-1β, IL-6, interferon-gamma (IFN-γ), and interleukin-17 (IL-17) are common findings in degenerating IVDs [59,114,127,128,129,130]. The effect of these elevated inflammatory molecules on IVDCs was investigated by Gabr et al. [130]. IVDCs from patients undergoing surgery for DD or scoliosis were stimulated with IL-17 in combination with IFN-γ or TNF-α. The stimulation of IVDCs significantly increased the secretion of inflammatory molecules (nitric oxide (NO), PGE2, IL-6, and intercellular adhesion molecule-1 (ICAM-1)), thereby indicating the potential of stimulated IVDCs to recruit immune cells to the IVD tissue [130].

### 3.3. Immunomodulatory vs. Regenerative Mode of Action in DDD

An ex vivo experimental study—using a bovine model of IVD degeneration—investigated the effect of a degenerative IVD microenvironment on the regenerative and immunomodulatory potential of human MSCs in co-culture with the bovine IVD [131]. No notable effect on ECM remodeling by MSCs was found, but evidence was presented for an immunomodulatory paracrine effect of MSCs, suggesting a predominant cytokine feedback loop between MSCs and disc cells [131]. In summary, the co-culture of human MSCs with bovine IVDs in an inflammatory environment led to the significant downregulation of bovine pro-inflammatory cytokines, including IL-6, IL-8, and TNF-α [131]. This study indicated that the immunomodulatory potential of MSCs might be more relevant than the regenerative capacity in an inflammatory IVD microenvironment. Furthermore, it raised the question as to what extent the regenerative capacity of MSCs contributed to the favorable outcomes of previous clinical trials on intradiscal MSC injections in DDD patients. In conclusion, the immunomodulatory potential of MSCs is an important factor in discal MSC therapy that deserves more attention.

## 4. Patient Stratification: MSC Therapy for DDD Patients with MC1

The clinical benefit of intradiscal MSC injections in clinical trials was inconsistent and had occasionally been reported to be restricted to a group of responders [1,132]. These inconsistent study outcomes can partially be explained by inconsistent study designs. Most of the clinical trials on MSC injections in DDD patients had low patient numbers and lacked a placebo control group and a standardized cell preparation [1,86,133]. Another important reason for the inconsistent clinical trial results might be the highly heterogeneous study populations. The multifactorial etiology of DDD makes it difficult to select a homogeneous and representative study population. Inclusion and exclusion criteria differed from trial to trial, thus leading to heterogeneous study populations and large inter-study variations. Stratification of patients into clinically meaningful subpopulations is essential for demonstrating the efficacy of MSC treatments in the highly heterogeneous DDD population. Selecting a cohort of potential responders based on clinical presentation could help to achieve consistently high therapeutic efficacy. DDD patients with MC are clinically different from DDD patients without MC in terms of pain perception and severity, standard treatment efficacy, and duration of LBP episodes [6,8,9,11,40,134]. Therefore, MC patients may represent a more homogeneous subpopulation of DDD patients and should be specifically investigated. Unfortunately, DDD patients with MC were not differentiated from DDD patients without MC in clinical trials on intradiscal MSC injections in DDD patients. In fact, patients with MC were occasionally excluded from the clinical trials, thereby possibly omitting a group of potential responders to MSC injections [73,75,135]. Therefore, the efficacy of discal MSC treatments against MC1 has yet to be investigated in future clinical trials.

The three different types of MC may represent different stages of the same pathology [25]. However, a stronger stratification might be beneficial. Assuming that the anti-inflammatory action of MSCs is a main contributor to the beneficial effects seen in the trials, it might be reasonable to focus on MC1. MC1 are the MC type with the strongest inflammation and with the highest pain association [13]. MC1 account for around 20% of all MC cases and appear as bone marrow edema-like changes in MRI, indicating areas of inflammation [15,136]. Investigating MC1 patients, as a clearly defined subpopulation of DDD patients, could significantly reduce study population heterogeneity. MRI could serve as a reliable technique to specifically include MC1 patients and to monitor treatment efficacy as a companion diagnostic. While infectious MC1 may be treated with antibiotics [36,37] and should be excluded from clinical trials on MSC injections, the main focus should be on the autoinflammatory MC1. Therefore, it is important to develop diagnostic tools to distinguish infectious from autoinflammatory MC1, enabling a causal treatment. The shape of the MC lesion seen on MRI has been shown to correlate with the infectious etiology of MC1. The absence of a ‘claw-sign’ of the MC1 lesion suggests an infectious etiology [137,138,139]. Magnetic resonance spectroscopy of the disc is another promising tool that may be able to identify MC1 related to infection of the disc with *Cutibacterium acnes* [140,141]. No study has reported a serum biomarker for MC1 that stratifies for infectious and autoinflammatory MC1, yet recent studies indicate different pathomechanisms in the bone marrow [142].

## 5. Possible Mode of Action of MSCs in MC1

To address the complex pathogenesis of MC1, a broad treatment approach targeting multiple aspects of the disease might be favorable. Multimodal MSCs seem ideal to address the vicious IVD/bone marrow cross-talk through immunomodulation of the inflammatory DD environment, regeneration of the degenerated IVD, and repair of the CEP through chondrogenic differentiation or stimulation of host repair responses [1,110]. However, MSCs have not been used to specifically treat CLBP patients with MC1. A major advantage of MSC-based treatment approaches over steroids, bisphosphonates, and TNF-α inhibitors is the long-acting multimodal action of MSCs. As MC1 is a multifactorial disease with characteristic CEP defects, DD, nerve ingrowth, thinned trabecular bone, and active vertebral inflammation, short-term immunosuppression, inhibition of TNF-α, or suppression of osteoclast activity alone might not be sufficient to alleviate CLBP.

Discal inflammation seems to play an important role in MC1 pathology, as the increased diffusion of inflammatory molecules from ‘MC1 discs’ [25,143] through the damaged endplates into the bone marrow might promote intensified inflammation in adjacent nutrient-rich vertebral bodies. Therefore, suppressing the inflammation in ‘MC1 discs’ could represent a valid treatment strategy to disrupt the inflammatory and pro-fibrotic feedback loop between the ‘MC1 disc’ and the bone marrow. The immunomodulatory cytokines TSG-6, IL-10, and TGF-β secreted by MSCs counteract the pro-inflammatory effects of TNF-α and IL-1β in IVDs. TNF-α and IL-1β are key inflammatory mediators found in DD [59] and are frequently used in in vitro experiments to investigate the effect of discal inflammation on IVDCs [133,134,135]. IL-1β stimulation of IVDCs induces the downregulation of stemness-associated genes and upregulates pro-inflammatory, pro-angiogenic, and catabolic genes [141,143,144]. Treating IL-1β-stimulated NPCs with TSG-6 reduced IL-6 and TNF-α secretion, increased the proliferation rate of NPCs, and promoted ECM synthesis [133]. IL-10 was found to increase the expression of ECM-associated genes and decrease the expression of inflammatory genes in IL-1β-stimulated NPCs [135]. TGF-β stimulation of degenerative AF cells grown in micromass culture increased the ECM production [145]. Moreover, TGF-β1 stimulation was shown to significantly increase the ECM production in NP cells isolated from degenerated human IVDs [146]. TGF-β3 treatment of degenerative human NP cells stimulated NP cell proliferation and induced an anti-catabolic gene expression profile, highlighting the regenerative potential of TGF-β [147]. In addition, co-culture experiments of degenerative IVDCs with MSC showed downregulation of IL-6 and IL-8 gene expression levels—pro-inflammatory cytokines that were found to be elevated in ‘MC1 discs’ [25,143]. Altogether, these studies provide compelling evidence that MSC can suppress discal inflammation. Moreover, suppression of discal inflammation might terminate nerve ingrowth into the IVD, as IL-1β was shown to stimulate the expression of vascular endothelial growth factor (VEGF), NGF, and brain-derived neurotrophic factor (BDNF) in degenerative IVDCs, resulting in angiogenesis and innervation [148]. Microvascular blood vessels expressing NGF were shown to enter painful IVDs from the adjacent bone marrow through the CEP [149]. These microvascular blood vessels were accompanied by neurotrophic receptor tyrosine kinase 1 (TrkA)-expressing nerve fibers. Thus, the suppression of discal inflammation might ultimately help to relieve discogenic pain.

The CEP is damaged in MC1, allowing a pro-inflammatory ‘MC1 disc’/bone marrow cross-talk. Repair of the leaky CEP might restore the IVD/bone marrow barrier, thereby disrupting the pro-inflammatory ‘MC1 disc’/bone marrow cross-talk. MSCs can promote cartilage regeneration by differentiating into chondrocytes to replace damaged cells or by promoting the proliferation and ECM deposition of resident chondrocytes through the secretion of cytokines, growth factors, and extracellular vesicles [89]. There are no studies on vertebral CEP regeneration. However, clinical trials on MSCs for cartilage repair in patients with cartilage degeneration—including osteoarthritis—have demonstrated encouraging results [150]. Single intra-articular administration of MSCs into the knee of patients suffering from degenerative joint disease or osteoarthritis showed cartilage maintenance or growth. Functionality of the joint was improved and pain relief was achieved in most patients [151,152,153]. CARTISTEM^®^—a composite of allogenic MSCs and hyaluronic acid—has already been approved by the Korean Ministry of Food and Drug Safety in 2012 for the treatment of cartilage degeneration including degenerative OA. A 7-year follow-up study demonstrated the persistence of the regenerated cartilage [144].

The trabecular bone is thinned in MC1, possibly due to osteoclast activity. Suppression of osteoclast activity might consolidate inflamed vertebral bodies, thereby improving the tolerance for mechanical load. MSCs support osteoclastogenesis under physiological conditions through the secretion of RANKL and M-CSF [154]. On the contrary, MSCs suppress osteoclast formation and activation under inflammatory conditions, through the secretion of osteoprotegerin (OPG) and IL-10 [155,156,157]. Therefore, MSCs might suppress the inflammatory trabecular bone resorption observed in MC1.

In summary, the regenerative and immunomodulatory properties of MSCs might counteract the painful molecular changes observed in DDD patients with MC1 (Figure 3). Firstly, regeneration of degenerated IVD and suppression of inflammatory bone resorption might restore normal functioning of the motion segment, thereby disrupting the degenerative cascade [78,80,158]. Secondly, CEP regeneration might repair the leaky IVD/bone marrow barrier, thereby disrupting the pro-inflammatory cross-talk observed in MC1. Thirdly, suppressing disc inflammation and the cross-talk with the adjacent bone marrow might suppress vertebral inflammation, nerve ingrowth, and ultimately discogenic pain [19,60,86].

We next discuss whether patients with MC1 should be treated with MSCs.

## 6. PRO—MSCs for Patients with MC1

MSCs represent a valid treatment option for patients with MC1 for several reasons. Firstly, patients with MC1 represent a homogeneous subpopulation of DDD patients with a clear inflammatory phenotype. Treatment approaches in the homogeneous patient populations can result in larger and more consistent treatment effects, as seen in clinical trials on intradiscal and epidural steroid injections in CLBP patients with MC [39,40,41,54,55]. Secondly, MSCs were shown to have a favorable safety profile, with no reported serious adverse events in completed clinical trials on intradiscal MSC injections in DDD patients [1,72,73]. Besides the absence of serious adverse events, DDD patients treated with MSCs experienced substantial pain relief and disability-related improvements [1]. Thirdly, MSCs can break the inflammatory ‘MC1 disc’/marrow cross-talk and heal MC. The multimodal mode of action of MSCs might play a crucial role in the treatment of MC1. MSCs are potent immunomodulators [125,126,141] with the potential to suppress the inflammation in the ‘MC1 disc’ and the cross-talk with the adjacent bone marrow. Suppression of the inflammatory and pro-fibrotic cross-talk could also suppress nerve ingrowth into the IVD, thereby suppressing discogenic pain [19,60,86]. The structural damage of the IVD and the CEP in MC1 could lead to altered biomechanics of the vertebral motion segment, which in turn could result in discal inflammation [76]. MSCs could target the underlying biomechanical issue by regenerating the degenerated IVD and the disrupted CEP, thereby alleviating or even preventing vertebral inflammation and subsequent degenerative processes. CEP regeneration might repair the leakage in the IVD/bone marrow barrier, thereby disrupting the inflammatory cross-talk between the ‘MC1 disc’ and the adjacent bone marrow. Furthermore, MSCs might suppress inflammatory trabecular bone resorption [155,156,157], thereby consolidating the inflamed vertebral bodies and improving the tolerance for mechanical load.

## 7. CONTRA—MSCs for Patients with MC1

A major concern regarding the use of MSCs for IVD repair is the harsh IVD microenvironment [77]. Although MSCs might counteract DD by chondrogenic differentiation, the differentiation of MSCs into chondrocytes is inhibited by IL-1β and TNF-α [159]—key inflammatory mediators found in DD [59]. This implies that IVD regeneration might be inhibited in the inflammatory ‘MC1 disc’. The number of viable MSCs needed to induce significant regeneration of the IVD is high and unlikely to be achieved in the hostile ‘MC1 disc’ [77]. Therefore, suppression of discal inflammation prior to the regeneration of the IVD and CEP might be a more promising treatment approach [159].

The positive outcomes in clinical trials on intradiscal MSC injection in DDD patients pointed to a regeneration-independent mode of action. Patients experienced substantial pain- and disability-related improvement despite the absence of discal regeneration [1,76]. However, if MSCs do not regenerate damaged IVD structures, the structural damages might worsen functional impairment and potentially lead to secondary pathologies of the motion segment.

As infectious MC1 may be treated with antibiotics [36,37] and should be excluded from clinical trials on MSC injections, the main focus should be on the autoinflammatory MC1. Unfortunately, no imaging or serum biomarker for MC1 exists that reliably differentiates between infectious and autoinflammatory MC1. Therefore, it is important to develop diagnostic tools to distinguish infectious from autoinflammatory MC1 that enable a causal treatment.

## 8. Conclusions

The efficacy of intradiscal MSC injections against MC1 has yet to be investigated in future clinical trials. The main goal of future trials on MC1 patients—as a homogeneous population with a clear inflammatory phenotype—should be to alleviate pain and slow down disease progression. These studies should focus on immunomodulation of the ‘MC1 discs’ and the repair of CEP damage to disrupt the inflammatory IVD/bone marrow cross-talk. MSCs might represent a safe and multimodal treatment approach with promising immunomodulatory and regenerative properties. Regeneration of structural damage to the IVD and CEP could prevent discal inflammation and possibly cure MC1 by disrupting the inflammatory IVD/bone marrow cross-talk. Advanced stages of DD might be an exclusion criterion for MSC injections, as the loss of structural integrity of the IVD and poor nutrient supply could impair the therapeutic efficacy of MSCs in this hostile IVD environment. No data on pH, osmolarity, and oxygen concentration in ‘MC1 discs’ are published, which might hamper the development of discal treatments. Assessing the condition of the CEP could be particularly important and should be considered when selecting patients for an MSC-based therapy. Therefore, novel tools to identify non-infectious MC1 and to assess CEP defects are required. Patient stratification, standardization of MSC preparation techniques, and selection of immunomodulation-related endpoints might pave the way for efficacious MC1 treatments.

## Figures and Tables

**Figure 1 ijms-23-02721-f001:**
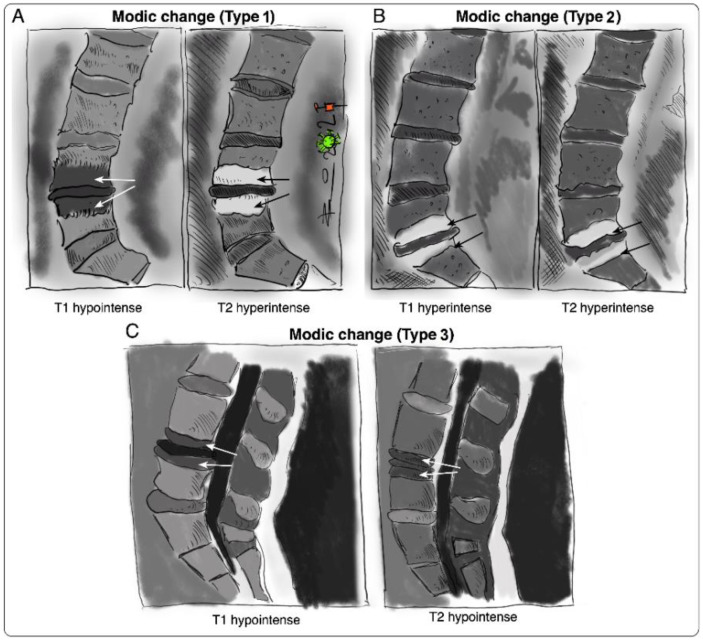
Sketches of intensity changes when scanning vertebral columns of human patients and classification of the three distinguishable MC according to T1- and T2-weighted sequences on MRI [15]. MC are classified into (**A**) MC type I, hypointense in T1 and hyperintense in T2, (**B**) MC type 2, hyperintense in T1 and T2, and (**C**) MC type 3, hypointense in T1 and T2.

**Figure 3 ijms-23-02721-f003:**
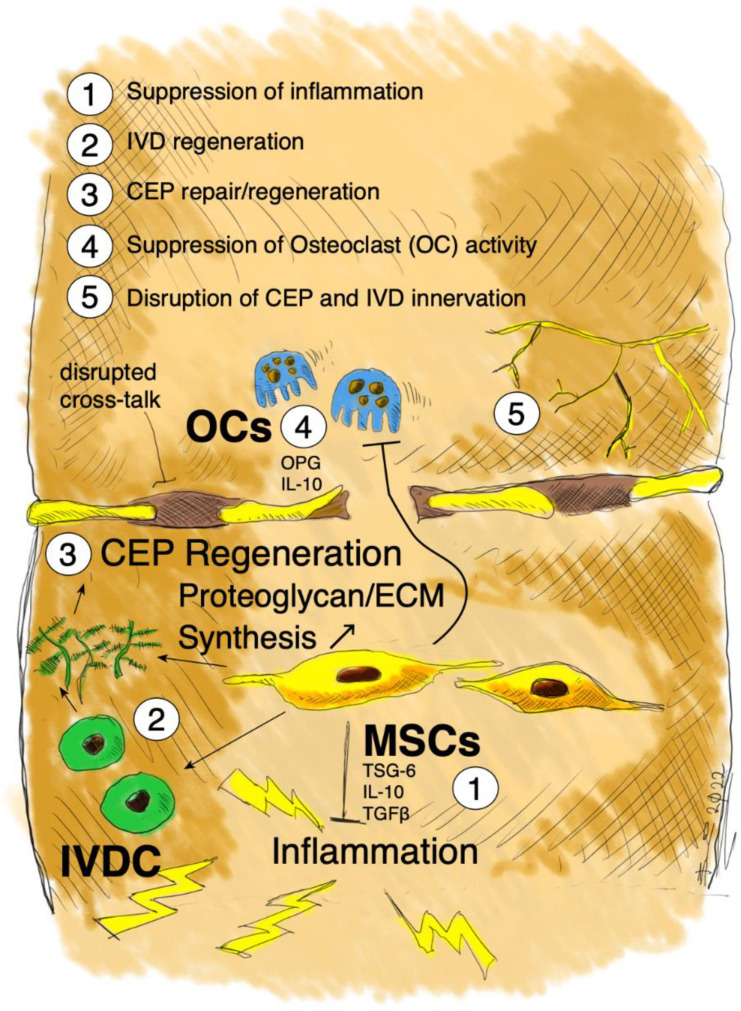
Schematic illustration of the possible multimodal mode of action of intradiscally injected MSCs in MC1. MSCs might regenerate the ‘MC1 disc’, repair the CEP leakage, and suppress osteoclast activity, thereby improving the tolerance for mechanical load. Suppression of discal inflammation and sealing of the CEP leakage might disrupt the inflammatory ‘MC1 disc’/bone marrow cross-talk, thereby suppressing nerve ingrowth and discogenic pain. OCs = osteoclasts.

**Table 1 ijms-23-02721-t001:** List of pro-inflammatory, pro-osteoclastic, and neurotrophic cytokines with elevated expression levels in ‘MC discs’.

MC Type	Pro-Inflammatory	Pro-Osteoclastic	Neurotrophic
MC1	CCL2, IL-6, IL-8, PGE2	OSCAR	NTRK1
MC2	CCL2, CXCL5, GM-CSF, IL-1β, M-CSF	RANKL, RUNX1, RUNX2	NTRK1

CCL2, C-C motif chemokine ligand 2; CXCL5, C-X-C motif chemokine ligand 5; GM-CSF, granulocyte-macrophage colony-stimulating factor; IL-1β, interleukin-1β; IL-6, interleukin-6; IL-8, interleukin-8; M-CSF, macrophage colony-stimulating factor; NTRK1, neurotrophic receptor tyrosine kinase 1; OSCAR, osteoclast-associated Ig-like receptor; PGE2, prostaglandin E2; RANKL, tumor necrosis factor superfamily member 11; RUNX1, runt-related transcription factor 1; RUNX2, runt-related transcription factor 2.

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
