# Peer review of "Should Degenerated Intervertebral Discs of Patients with Modic Type 1 Changes Be Treated with Mesenchymal Stem Cells?"

_ijms, 2022, doi:10.3390/ijms23052721_

Round 1

Reviewer 1 Report

In this review, authors discussed whether degenerated IVDs of patients with Modic type 1 changes should be treated with an intradiscal MSC injection and summarized outcomes from anti-inflammatory clinical interventions in discogenic LBP patients. And I do have several suggestions for improving the manuscript:

  1. To make it more readable in the abstract and conclusion. Especially, the conclusion needs to be express concisely.
  2. Page 3, line 115. “Therefore, inhibiting disc inflammation may have the potential to break the inflammatory disc/marrow cross-talk and heal MC.” The sentence needs to be reorganized.
  3. Please strengthen your writing ability and professional terminology. Need grammatical and typographic check.
  4. Why choose MSCs.
  5. Please add a new part to summarize the shortcomings of the literature at present and prospective. For a review, it is vital to readers to get new information, not simple listing other researches.
  6. The picture quality should be improved.
  7. Page 15, line 566. "Therefore, the efficacy of discal MSC treatments against MC1 remains unknown." The opinion need be put forward in abstract and summarizing paragraphs.

Author Response

Point 1: To make it more readable in the abstract and conclusion. Especially, the conclusion needs to be express concisely.

Response 1: We improved the readability of the abstract and the conclusion. The conclusion was written more consicely to focus on the most important points discussed in this review and to address the question if IVDs of patients with MC1 should be treated with MSCs.

Point 2: Page 3, line 115. “Therefore, inhibiting disc inflammation may have the potential to break the inflammatory disc/marrow cross-talk and heal MC.” The sentence needs to be reorganized.

Response 2: The sentence was reorganized to better summarize the paragraph.

Point 3: Please strengthen your writing ability and professional terminology. Need grammatical and typographic check.

Response 3: We checked the manuscript for spelling mistakes and rephrased sentences that were formulated unclearly. Additionally, the manuscript was checked by a native English-speaking colleague to correct grammatical and typographic mistakes.

Point 4: Why choose MSCs.

Response 4: Many thanks for pointing out that the discussion if MSCs are suitable for the treatment of 'MC discs' was not clear. This is in fact a central point of this manuscript.

In chapters 5-7, we aimed to provide a discussion if MSCs – which are widely used in clinical trials on DDD – should be used to specifically treat MC1 patients. We compared the multimodal mode of action of MSCs with the pathologic bone marrow changes in MC1 to discuss if patients with MC1 should be treated with MSCs. We described that MSCs might target multiple pathologic molecular and cellular changes found in MC1, for instance suppression of discal inflammation, regeneration of the ‘MC disc’ and the damaged CEP, suppression of inflammatory trabecular bone resorption, and finally disruption of the CEP and ‘MC disc’ innervation. Short-term immunosuppression, inhibition of TNF‑α, or suppression of osteoclast activity alone might not be sufficient to alleviate CLBP in MC1 patients as seen in clinical trials on glucocorticoid injections, TNF‑α inhibitors, and bisphosphonates, respectively. Long-acting multimodal MSCs might show a larger treatment effect.

We structured the discussion if MSCs should be chosen for the treatment of MC1 in PRO and CONTRA, to not only highlight the clinical potential of MSCs but also to review challenges of intradiscal MSC injections. A major challenge of intradiscal MSC injections is the harsh IVD microenvironment, which likely limits MSC survival and function. MSC licencing strategies and biomaterial scaffolds might overcome this hurdle by improving the survivability and the therapeutic potential of MSCs in the harsh IVD microenvironment. We concluded, that MC1 patients should be considered for intradiscal MSC injections.

We implemented a few sentences in the conclusion to summarize if an intradiscal MSC injection in patients with MC1 could be beneficial.

Point 5: Please add a new part to summarize the shortcomings of the literature at present and prospective. For a review, it is vital to readers to get new information, not simple listing other researches.

Response 5: The reviewer points out an important point. The original manuscript did not clearly summarize the shortcomings of the literature in the context of the question if patients with MC1 should be treated with intradiscal MSCs. We added a new part in the conclusion to make the readers aware of important gaps in the literature.

Point 6: The picture quality should be improved.

Response 6: We appoligize for the low picture resolution. Microsoft Word automatically compressed the pictures. Were re-uploaded the pictures with higher resolution.

Point 7: Page 15, line 566. "Therefore, the efficacy of discal MSC treatments against MC1 remains unknown." The opinion need be put forward in abstract and summarizing paragraphs.

Response 7:

We reformulated the sentence and moved it to chapter 4. The core statement that there is no published information on the efficacy of discal MSC treatments against MC1 was revisited in the conclusion to make the readers aware of shortcomings in the literature.

Reviewer 2 Report

The present manuscript titled " Should degenerated intervertebral discs of patients with Modic type 1 changes be treated with mesenchymal stem cells?" reviews an important topic of use of MSCs for treatment of degenerated disc disease with Modic type 1 changes. The manuscript has done a thorough review of the existing literature in an excellent manner with an appreciated flow of information for readers.  The manuscript can be accepted in the current form.

Author Response

Dear reviewer 2, many thanks for your positive feedback. We corrected minor spelling mistakes and rephrased a few sentences that were difficult to understand. Additionally, the manuscript was checked by a native English-speaking colleague to correct grammatical and typographic mistakes. Furthermore, we implemented several improvement suggestions of reviewer 1. We restructured the conclusion to improve readability and to make it more concise. We added a new part to highlight the shortcomings and gaps of the literature at present as recommended by reviewer 1. Finally, we found a small (but important) error in Figure 2. Bacteria were drawn in the MC bone marrow, however, they are restricted to the disc tissue. We moved the bacteria from the MC bone marrow to the disc tissue.